# “Sports for All”—An Evaluation of a Community Based Physical Activity Program on the Access to Mainstream Sport for Children with Intellectual Disability

**DOI:** 10.3390/ijerph191811540

**Published:** 2022-09-14

**Authors:** Florian Pochstein

**Affiliations:** Faculty of Special Needs Education, University of Education Ludwigsburg, 71634 Ludwigsburg, Germany; pochstein@ph-ludwigsburg.de

**Keywords:** physical activity, intellectual disability, mainstream sport, perceived barriers

## Abstract

Access to club sports is still not a given for children with ID. Parents and children report numerous structural and social barriers to accessing mainstream sports. Sports clubs, on the other hand, want to include this group of people, but often do not know how to do it. Using a community-based approach, children with intellectual disabilities (8–15 years) and their parents were given the opportunity to participate in an 8-week sports program in four mainstream clubs organized by a self-help organization for people with intellectual disabilities. Focus groups were conducted with all participants (parents, children, and club representatives) before and after the program and evaluated by means of a thematic analysis. The children rated participation very positively and only very occasionally reported that they had been excluded. The parents confirmed this experience, but were nevertheless more critical in their assessment. Outside the research context of this study, the sports clubs hardly provided good support, which can also be deduced from previous negative experiences of the parents. The sports clubs themselves received valuable information about what their failings were: sufficiently available and qualified staff, better access to sports facilities, and a wider range of inclusive groups are needed. On the other hand, there was hardly any prejudice on the part of the non-disabled club members towards the new members with ID, which was evaluated extremely positively, albeit surprisingly, by the sports clubs.

## 1. Introduction

Regular physical activity (PA) is an important determinant of health benefits for all. The positive effects of PA on a wide range of physical, social, and mental health are well known and widely recognized [1]. Numerous qualitative studies have also demonstrated these benefits for people with intellectual disabilities (ID) [2,3,4]. In addition to these studies, recent meta-analyses have shown moderate to large effect sizes for PA in improving physical [5] and mental [6] health.

Unfortunately, people with disabilities are the group with the lowest regular participation in PA [7,8,9]. A meta-analysis [10] confirmed that children with ID participate in less PA than children without ID, and a recent systematic review found that only 9% of adults with ID meet the minimum requirement of 150 min of moderate to vigorous PA per week [11]. Compared with children without disabilities, peers with developmental disabilities have significantly lower levels of PA and are less likely to meet the proposed guidelines [12,13]. When considering individuals with severe disabilities (i.e., high to very high support needs), participation rates drop tremendously [14]. However, even for people with mild to moderate ID, participation in sport still is not yet a matter of course, and only a very small number participated in organized sport, like clubs [15] or public facilities (e.g., gyms) [16]. Nevertheless, this kind of sport seems to have better effects on social and psychological well-being as well as self-perceived inclusion compared to clinic-based settings or therapeutic offers which are more common for this population [17].

Only a very few studies [18] focus on the barriers (and facilitators) to participate in organized sport for people with ID. Most of this research has focused on the parents’ point of view as they play a crucial role in initiating and reinforcing PA for their children [19,20]. Even when the children grow older, parents are the deciding factor in supporting the transitions (e.g., from school to work or from separated to inclusive sport) of their children [21] and, generally, parental support has long been noted as being crucial for the participation in PA among children with (and without) disabilities [22]. A major finding of a systematic review [23] suggests that parents see multiple motives to engage their children in PA, like health and psychological benefits as well as the opportunity to socialize with other children or gaining more self-confidence and independence. On the other hand, a lack of program, a lack of qualified professionals, and a lack of parental skills to support the children in PA were reported as barriers.

People with ID are interviewed less frequently on the motives and barriers to participation in sport. One of the few studies on this subject [24] showed that adolescents with ID seemed to be intrinsically motivated to participate in PA and organized sport. Support from significant others (like parents, teachers, and friends) tends to be crucial as well. Lack of awareness for the special needs of people with ID among providers of PA and society in general and the fear of being excluded and not accepted were also cited.

Participation in organized sport can be beneficial for social, educational, cultural, and health reasons [25], but a recent review could not identify much evidence for the assumption that participation in sports programs improved community participation and inclusion for people with ID outside the program, although the development of friendship during the programs was demonstrable [26]. Personal, social, and environmental barriers and barriers at the policy level have been identified as impediments [4,27], but little is known about how community sport clubs include people with ID in their activities and the impact on people with (and without) ID [19,28].

For the reasons outlined above, the following questions guided the present study.

What are parents’ positive and negative experiences with their children’s participation in organized sports? What do they expect? What are they afraid of?What do the children with ID expect? What positive and negative aspects can be identified?What is the view of the sport clubs? What conditions must be met for them to open up to people with ID? What benefits and what responsibility do they see in this area?What are the implications for all stakeholders if people with ID participate in organized sports?

## 2. Materials and Methods

### 2.1. Participants

Fifteen families with children with ID took part in the study. They all were recruited during a community-based approach on participation in PA organized by a local agency of the German Self-Help for people with ID and their families [29]. The agency offered the actual sports program on their website, stating that participation is open for everyone with the aim to find ways to get into mainstream sports for children with ID. Families had two months to register. After this, 15 interested families got an information letter explaining the participating clubs, the sport offered, the timetables and addresses of the clubs, and the specific aims of the accompanying study. The families consisted of parents and children with ID and siblings without ID. From the 15 families, 10 families took part with mothers and fathers, while five families only attended with mothers. The age of the 15 children with ID ranged from 8 to 15 years (M = 11.47; SD = 2.52); seven families had siblings without disabilities in an age range from 3 to 22 (M = 15.56; SD = 5.78). The participating children were reportedly diagnosed with the following disabilities: down syndrome (N = 7), cerebral palsy (N = 3), autism spectrum disorders (N = 2), attention deficit and hyperactivity disorder (N = 2), and developmental delay (N = 1). All information about the diagnoses were given by the parents. All families described themselves as well organized and very interested in supporting their children to live their lives as normally as possible. Inclusion in society was a central aim of all families; for this, the participation in this program was reported as another way of reaching this aim.

The four mainstream clubs offered a various array of sports, including team sports (for instance, soccer, handball, and basketball) and individual sports (athletics, gymnastics, and swimming, among others) in the near community. All clubs did not have special sport groups for people with ID nor other disabilities but were basically open for all and felt responsible for social inclusion, as stated on their websites.

Ten female students from the local university studying special needs education, supported the families and the clubs during the program as facilitators. All students had long experience in mainstream sports (as athletes and/or coaches) as well as in supporting children with special needs (ID and other disabilities). The rationale for engaging them was not just for practical purposes but was also built on the approaches known to be helpful in creating inclusive settings such as in supported employment or in schools. They were not intended to act as a personal coach or a caretaker, but to foster an inclusive and equal participation of all members of the sport clubs.

### 2.2. Sports Program Structure

The Self-Help agency offered an eight-weeks sports program in which the participating families could choose one or more types of sports within the offer of the four cooperating sport clubs. They were open to try different opportunities with the aim of finding one or more types of sport which fit with the demands of the children with ID and/or the family as a whole. During the eight weeks, most of the children found a suitable sport. The sports were chosen by interest and availability in the clubs and covered handball, gymnastics, swimming, equestrian, and soccer.

Two families dropped out after three weeks of participating. One family reported problems in getting their child to the clubs. The distance from the living area to the sport facilities was too far, so the effort was not manageable. The child of the second family indicated that she was bullied by the non-disabled kids on the team. Although parents, the coach, the facilitating student, and the author had serious discussions with the kids, no solution was found to convince the girl with ID to continue her participation. She did not want to continue in this group and was not motivated to try again in another group.

Supplementary to the sports program, the families and the sport clubs representatives took part in two rounds of focus groups, one at the beginning of the program in week one (t1) and another at the end of the program, after week eight (t2). The complete process of the study is shown in Table 1.

### 2.3. Data Collection

Focus group interviews were chosen because this method enables a process of sharing ideas and comparing them between all persons interviewed in an interactive way, thereby leading to a co-constructing of meaning by the participants [30]. At t1 and t2, there was one focus group for the children, one for the parents, and one for the club representatives, all lead by the author and two trained students, who were not taking part in the sport program to ensure a conducive environment for speaking freely about their experiences, thoughts, hopes, and fears.

An ethical approval for the study was obtained by the local university. Information sheets about the evaluation aims and consent forms were given to all participants, using an easy language format for the children with ID. For the children who were not able to read, the documents were read to them by the assisting students, who were familiar in interacting with children with ID.

After obtaining consent forms and introducing the purpose of the program evaluation, the interviewers used semi-structured interview guides with predetermined topics (see Table 2). Despite this fundamental structure, the order and phrasing of the underlying questions stayed open and flexible for adjustments in order to follow the way of thought of the interviewees. All interviews were audio-recorded, and lasted from around 18 min (for children with ID) to 32 min (club representatives) and 47 min (parents) at t1.

The second focus group round at t2 lasted around 12 min (children with ID), 21 min (club representatives), and 26 min (parents) and had the aim of summarizing the feelings and experiences after the eight-week program. For this purpose, the questions were adjusted as seen in Table 3.

### 2.4. Data Analysis

The focus group sessions were audio-recorded and verbatim transcriptions were made. As the main goal of the study was to search for patterns across the different interviews rather than within a single case, thematic analysis [31] was used to analyze the interviews. Thematic analysis tries to generate overarching themes of the constructed reality of the participants by coding, categorizing, and drawing thematic maps. In an iterative process, the author and the two trained students went through the data set several times to finally come to a consensus about the underlying themes and codes [32].

## 3. Results

The thematic analysis revealed several themes, which were related to the experiences, expectations, and wishes from all stakeholders. The following section will describe these themes broken down by time (before (t1) and after (t2) the program) and group (parents, children with ID, sports club representatives). The final thematic maps with main themes for all stakeholders can be seen in the Appendix A.

### 3.1. Parent-Reported Themes before the Program

All parents reported a great importance of physical activity for their kids. The perceived child’s need for motor skills training and psychosocial support was the main motivator to join the program. In contrast to motor therapies, which most of the children went through several times in their life, sport offers better possibilities for “real life” training together with other kids. A mother described these aspects in the following words: “I mean, for him [8-year-old son], it’s so important to train his body. He does not move much in his life. I wish him to be more active and have fun while doing so. In his therapy he is more passively moved than acting by his own. I hope, that he will enjoy the sport activities together with the other kids in his age”.

The former experiences with taking part in mainstream sport were very negative. Out of the 15 families taking part in the first focus group, 10 had tried to be active in a mainstream sport club, some for several times. The anticipated positive aspects motivated the parents to try a membership in the clubs, but most of them did not feel welcomed and reported of unprepared club structures, not qualified coaches, and—sometimes—unwillingness to include their kids in the running of the programs. A father stated: “At the beginning, all went quite good in the club. I am a member there for five years and all knew about my daughter, and this was never a problem, as long as she was not participating. But when she started to join the regular swimming program, the situation changed rapidly. The other kids were not very kind, some of them were mocking about her, some seemed to have fear or just were overwhelmed. That is ok for me, they are young kids. But I expected the coach to act. She didn’t do. She just wasn’t able to handle the situation, which really disappointed me”. A mother of a 12-year-old girl added: “That’s exactly the reason why we are here. We tried to include her in the local soccer team, she loves kicking. But the coach didn’t know how to handle her. She needs a little more time, ok, but she is able to follow the rules and tries to give her best. He [the coach] was not able to adapt his training to her abilities”.

On the other hand, there were some positive aspects as well, especially concerning the attitudes of the other, non-disabled children in the sport groups. “My son felt very well in his handball team. He was the only member with ID, but the other kids didn’t mind. They treated him like all the others, and the coach was very engaged with learning how to adapt the training. All went well, until they get into competition mode. My son wasn’t good enough to compete against the other teams. When it came to win or lose, there was suddenly no place for him. He didn’t get the ball any more, was just standing at the edge of the playing field and lost the motivation to take part”. The last argument of a mother reveals an important topic. Even when participation in a regular sport program is working, which is rare enough, competition seems to be a fundamental problem. Including people with ID in a system where performance is the limiting factor seems to be a contradiction. A lot of sport groups take part in some kind of competitive league, and there are only a very small number of inclusive teams at this level.

The arguments stated echo reports in the literature [33]. Mainstream sport is often not accessible for people with ID with numerous barriers (e.g., lack of program, lack of qualified coaches, lack of understanding people with ID). Most of the parents had negative experiences with mainstream clubs, but still had the wish to enable participation in regular sport groups with the anticipated benefits on the physical and psycho-social level; this motivated the parents to search for other possibilities. One possibility was the present sports program. The expectations were huge, as for some families, the program seemed to be a kind of last chance to get into the sports system with their kids. “For us, this program sounds like the answer to all our problems. A lot of different sports, inclusive groups where our daughter may find friends without disabilities and has the opportunity of doing sports the way she likes and is able to. I really hope this will work. But I am unsecure about it. We tried so often, and it was a disaster every single time. What makes me confident is that it is accompanied by the Self-Help agency”. These statements of a father were affirmed by a mother of a 10-year-old girl: “I am very happy that we have the possibility to speak with the club representatives. I think they often just don’t know about the special needs of our kids and how easy it can be to fulfill them. But they are here and they seem to be very interested in offering the program and in having our kids participating in their clubs. This makes me very happy”.

### 3.2. Child-Reported Themes before the Program

The 15 children with ID discussed the same topics as the parents. The answers were shorter but the general direction was comparable. The importance of sports was presented very impressively. Being active, training muscles, and getting in good shape were motivators on the physical side, along with the aspect of learning new skills and general health. “I love to do sports. I can strengthen my muscles during weight-lifting or get more stamina through running. It’s very healthy as well”.

Despite being intrinsically motivated to do sports, support from significant others, like family, friends, and teachers, seems to be crucial in starting to be active for children with ID. This leads to the second topic of the focus group, the motivation to join the program. Some of the children did not really know what the program was about, but some were very well informed and gave their comments on it. A 15-year-old boy summarized it as follows: “My parents asked me if I want to try it. It’s about joining normal sport groups in normal clubs. Not only with other kids with limitations. I really would like to do it. I want to play handball, and that is not possible where I live, there is no club for me. And I want to find new friends in the club. Maybe there is someone who is living nearby”. The aim of finding friends, of getting into social contact, was often mentioned by the children (as with the parents). This touched on another motivation for doing sports in general. Besides the physical aspect (see above), the psycho-social benefits of doing things together and the wish of finding friends and getting into contact with kids of the same age without disabilities was very pronounced.

Approximately half of the children (N = 7) talked about previous experiences with mainstream sport. The experiences were very similar to the statements of the parents. Even though some had positive memories of acting together in a team, most remember negative aspects like being left out or even bullied. The contact to the coaches was reported as insufficient. A girl of 11 years remembered her time in an equestrian group: “They laughed about me from the beginning. I was not as thin as the other girls and I was not as good as them. The coach was nice first but later on she asked me to look for another sport. My mum got into dispute with her and we never came back”. The children had very few specific wishes for the participation in the program. They were curious about the different sport groups and were hoping that the kids without disabilities were nice. All in all, it seems like the parents were the driving factor behind the participation of the children.

### 3.3. Clubs-Reported Themes before the Program

The motivation of the participating children and parents is one important aspect in creating an inclusive sport setting. At least equally important is the view of the providers of the sport programs. The motivation to join the project was centered around social responsibility and the wish to foster inclusion and curiosity. The aim to promote physical activity for all, including people with ID, was seen as a central task for sport clubs. With their longtime experience in organizing physical activities for young and older people, sport clubs see themselves as an ideal player to enlarge this to people with ID.

Another big issue was seen in promoting contact between children with and without disabilities as a central mission of inclusion. All these aspects were affirmed by the representatives, but this general attitude is not new of course. The concrete implementation and the ability to act as an inclusive club are considered desirable but the “know how” is missing. One interview partner admitted it like this: “I am very happy that [the author] asked us to take part in this project. I am thinking about including children with disabilities in our cubs for a long time. For me, this is a very important task. All areas of life should be open for everyone. But I have to be honest: I don’t really know, how this should work. I hope, we find ways in this project to open up our groups and our minds”.

The clubs all had some experience with people with disabilities in their clubs, but mostly this concentrated on family members who join with their non-disabled parents or siblings more or less regularly. A special or adapted program did not exist, nor any inclusive groups with qualified coaches. So, not much was said about former experiences, but a lot of aspects were articulated when it came to expected barriers of the participation of the children with ID. In summary, there were two main concerns. One was the expectation that the coaches were not ready for leading inclusive groups. Some may have had a general fear or a negative attitude against people with ID, but this was seen as a minor problem, because the clubs are looking for personnel who are more open minded and positive.

However, being open minded is not sufficient. The coaches need to know how to design activities that serve all participants. This is challenging, and becomes even more challenging if the group is very diverse. Being adequately trained and motivated is indispensable to offer a good program. All coaches work on a voluntary basis, so it is hard to put some pressure on them to qualify more and more. In addition, there is the problem that qualification programs are quite rare. The clubs do not really know about such programs, even if they do exist. However, they are mostly offered by the Disabled Sports Association, which is, at least in Germany, not so well connected to the general sports system. There is insufficient mutual exchange between the two systems.

All clubs see these problems, but they all have positive expectations as well. They see the benefit of offering possibilities for social contact, physical activity, competition, and fun. Further, not to forget, there is a financial aspect as well. The acquisition of new members is an important factor, as the following statement shows: “Honestly, we have the same problems like everyone. Our members get older, and we have difficulties to recruit younger persons. Becoming a member in a sports club is not as attractive for the kids any more as it was 20 years ago. We have to look for new groups, we have to become more attractive, especially for young people. For me, inclusive groups are a big chance to increase the number of members. We excluded people with disabilities much too long, it’s time to change it. Not only for getting new members of course, but this is an important aspect as well. We can’t offer such a variety of programs without an adequate number of members”.

### 3.4. Parent-Reported Themes after the Program

Thirteen out of fifteen families stayed in the program for eight weeks. The children with ID took part in several activities, mostly finding one favorable sport program each. The parents’ satisfaction with the program was on a very high level. In the second focus group, the parents discussed the initial challenges and the solutions found by the program. Two main aspects were especially mentioned. The improvement in the motoric and psychological abilities of the children was the most important topic for them. A mother said: “I was skeptical at the beginning, I have to say. I was not sure if he [her 13-year-old son] would manage it. I was afraid that it could be too exhausting to train for a whole hour every week. But that was no problem at all. He improved so much in everything. Mostly his motivation to be active is much bigger now, that is great”.

The second main aspect was the assistance of the students. They served as facilitators for both the families and the clubs. In the perception of the parents, this was the decisive point for the success of the whole program. On the one hand, this speaks for the program, but on the other hand, it shows which problems still occur, even in this structured setting. The clubs alone, with their existing structures and coaches, were not seen as qualified enough for meeting the demands of the children and the families. A father’s opinion shows this: “We as a family are very happy with the program. Our daughter was happy with her gymnastic group, all kids were very kind, the activity was well adapted and the coach was very affectionate. But it would have not worked without K [a student]. She was our fixed point. If there were any questions, she was there for our daughter and for us. She was the link between us and the club. She was there when we came to the club, she was there when we pick our daughter up after sport and gave us the feeling of being welcomed. For our daughter it is very important to have a permanent contact person. The coach can’t do this for all kids, her duty is in offering a good program, and she did. But all the little things around are so important as well”.

Some parents argued that they would like to have some more feedback from the coaches on the development of their kids in the group. A potential improvement for the future was seen in a kind of parents’ education. Questions arose about how to transfer an active life in to the whole family and make physical activity a permanent part of their child’s life. Only four families signed membership forms to stay in the clubs for a longer time. The reasons were different, but two aspects were central. One was due to the program structure, the other more a logistic challenge. The present program was embedded in a university study, and this was noticed by the parents, of course, like the following statement shows: “I mean, it was a very good experience for our son and us. But it is a kind of artificial situation? All the supporting students and structures are fantastic. This won’t be there in the future, so we decided not to become new members in the club for now. He needs the support of this kind of facilitation. If they [the clubs] manage this, we will be back again”.

The logistic aspect deals with a widely known problem. Getting to the clubs was not very easy for all families. The clubs are located around the city, but for some families, this meant long commutes. As the children are not able to use public transport on their own, the parents have to act as support persons. This was not manageable for some families. During the 8-week program, assistance was given by the facilitators, but as long as there are no persons responsible for this, this problem will be present.

### 3.5. Child-Reported Themes after the Program

The children mostly focused on positive experiences during the program. The possibility to act in their favorable sports was very well appreciated. Eight of thirteen spoke about the fun they had in doing sports, and more than half of them reported that they found friends without disabilities in the groups. Only two children said that they had met these new friends outside the training.

When asked about the most important aspect of the program, answers like “the variety of sports” [boy, 8 years old], “contact with normal kids” [girl, 10 years old], “chance to train my body” [boy, 15 years old], and “the students” [girl, 14 years old] were given. Negative aspects were mentioned more seldomly and concentrated on transport issues (comparable to the parents’ view) and, in two cases, on feelings of not being welcomed. A 9-year-old girl said: “Mostly I liked it. But in my group was another girl, who was mean to me. She didn’t want to play with me and said bad words. But the other kids and the coach were nice and told her to stop it”. Most of the children did not think about the time after the program. They did not care about membership or something similar. The short duration of the program was not clear for them, nor was the need to become a full member of the club to continue the sports program.

### 3.6. Clubs-Reported Themes after the Program

The feedback of the clubs was a central, and new, aspect of the whole study, as not much is known about their experiences. All clubs were not familiar with this kind of inclusive settings but all saw the need and the responsibility to offer such groups. The eight-week program was basically judged very positively. All club representatives reported about positive feedback from most of their members and the coaches. The inclusion of the children with ID worked quite well in most of the cases, which was surprising for the clubs. A female club president said: “At the beginning, I was not sure if this will work. But I am surprised how well it worked. I think, the kids with ID had lots of fun, as well as our previous members. The atmosphere in the groups was very nice, and you could see what is so unique about doing sports: people with different abilities perform together on the same subject. Some are more capable than others, but that’s ok for all. In my opinion the team spirit was great”.

However, there were some critical aspects as well, as can be seen from the following quote from a male general secretary of another club: “All in all I am happy, that we took part in the program. We just weren’t aware of this population I have to admit. It’s an absolute win for all, for the kids and for us as clubs as well. I really hope, this will work in the future. But there are still some aspects that I am concerned about. Some of the kids really need a lot of assistance. That worked well with the students, but without them? I can’t imagine, how the coaches or the other members of the group can handle this”. Again, the special role of the facilitators emerges. Without proper assistance, the inclusive groups will face several barriers, beginning with issues in the locker room and continuing with practical help during the physical activity. The given structure with one coach per group is not sufficient to meet the demands of the kids with ID nor the ones without in inclusive groups.

The coaches gave their feedback to the club representatives in a similar manner. In their opinion, inclusive groups could only work when at least two prerequisites were fulfilled. First of all, the coaches asked for more qualification concerning disability and inclusion-orientated topics. The curriculum of the existing training of coaches does not cover this at all. Second, there have to be at least two persons in each group, preferably at least one with experience in people with special needs.

The last big theme, which appeared from the thematic analysis, was the perception of the children without disability who trained together with the children with ID. At the beginning, the clubs were afraid that the previous members would have concerns about the inclusive group. Luckily, this was not the case. The sense of cohesion was quite high in all the groups, even if there was some mocking from time to time. It seems like the contact in the groups was perceived as a mutual favorable interaction, which is indispensable as is long known from contact theory [34]. Doing sports together with the same goals and values seems to be a way to foster mutual understanding, which can lead to better attitudes and behavior. The CEO of the biggest participating club has met all the coaches of his club and stated their opinions: “They told me, that there was nearly no problem between the kids. Hey, they were there to have fun, to sweat and to play handball and so on. Why should there be a problem. Some little disputes occurred, but this is normal, isn’t it? They are kids, they try their boundaries, nothing serious, and nothing which is not happening in non-inclusive groups. Maybe all kids, which anticipate problems just didn’t come for the eight weeks, but I don’t think so. If at all, then some parents may have problems’.

All four club representatives want to continue to open their groups for all. Officially they already are, but practically, only a few persons with disability found their way into the clubs. Through participating in this program, the clubs were reinforced in investing more effort in member recruitment and promotion for this group. “We have to do more, the potential is huge. Maybe we should build cooperation with special services or special schools” was the idea of one of the interviewees.

## 4. Discussion

Our findings replicate the results of several earlier studies [23,33]. Parents play a crucial role for children’s behavior anyway, but especially, children with ID are often dependent on their parents to be able to participate in leisure activities, like sport [35]. Parents are aware of this and all parents in our study tried to assist their children in getting access to mainstream sport as it is seen as a very beneficial activity. However, our results affirm the insight that this is hard to achieve. Structural barriers like inadequate facilities, lack of sport opportunities, or lack of qualified coaches exacerbate the difficulties that people with ID face because of individual aspects like problems in physical, cognitive, or social skills. This often leads to problems in inclusive groups, which in the end means retirement from sport. Only very few parents (and children) are motivated and strong enough to face these problems and try to find a solution, which makes is possible for the child to stay in the group as a fully accepted member. Parents know about the special needs of their kids and therefore much hope was placed in the present program, even though the expectations were somehow low for some parents.

That was different for the children. All were very excited and full of hope that participation in the program would work well. However, one has to take into account that half of the kids were not really informed about the content of the program, which again shows the dependency on the parents. Their experiences from previous mainstream sport activities were mixed with a focus on negative memories. Social interaction in team sports seemed to be especially critical, which confirms studies which have shown that inclusion is easier to reach in individual sports [36,37]. Nonetheless, most of the children chose a team sport in the preset program. It seems like they behave the same way, like their non-disabled peers: team sport is more attractive for children (at least in this age) because of the possibility to make friends inside the team [24]. Another critical aspect was the distinction between competitive and non-competitive sports. Geidne and Jerlinder [19] found that inclusion works better in non-competitive sports, because the will to win and the fear of losing a game often leads to the exclusion of persons with lower abilities. Again, this is the same for people without disabilities, and in the present sports programs, there were no competitive arrangements.

The clubs’ points of view provided insights on a little explored topic. Mainstream clubs play an important role in offering activities for all. In Germany, sport clubs are committed to offer equal participation for all people and break down barriers that prevent participation. They stand for an inclusive sports community that excludes no one. For this, they receive public financial support and are obliged to transform their programs into an inclusive direction. Up to now, this is mostly not fulfilled. Inclusive groups are still rare, and the distinction between a disabled and a non-disabled sports system is historically grown and still current [38]. However, society is, of course, moving on. Clubs try to include people with ID (and other disabilities) because they feel responsible and see the positive impact for all participants. The potential membership fees of the new members are of course also interesting, as the number of members decreases from year to year [39]. The four participating clubs showed very high interest and will to engage in this field. However, the fears are manifold. They are aware that they are not well positioned in the area of adapted sport offerings, in accessibility of the facilities, and in the qualification of the coaches. All these aspects are modifiable, but this will need effort, money, and, most of all, time. Nevertheless, they wanted to start now and volunteered to participate in the program of the Self-Help agency, even if the exact procedures remained somehow unclear for them.

Overall, all stakeholders were very pleased to have participated. The majority of the children had great fun, parents were pleased with the improvements of and the care for the children, and the clubs gained worthful experiences for their future development. However, a more in-depth analysis of the statements also revealed critical aspects. The children for instance reported about the new friends they found in the groups. This positive aspect was somewhat diminished by the fact that these friendships were exclusively related to the time spent in the sports groups. This finding is in line with recent studies, which showed that the transition of positive interactivity into daily life is only rarely observable [40,41]. On the one hand, you can argue that the overlap in real life still is not very pronounced and that exclusion still is more the reality than inclusion. However, on the other hand, it could be admitted that sport has high potential for social inclusion [42] and a normalizing function that could increase the quality of life [43]. It is definitely necessary to work further on the promotion of social interaction on the field and trying to support the transition to life outside the sport context.

Besides all positive aspects, parents focused on one central aspect, which was, in their perception, the core element that has made the program successful. They felt the student facilitators did a great job, as they were troubleshooters, caretakers, coach-assistants, and much more. The parents were overwhelmed by the effort the students put in. This is how an inclusive offer can work, maybe must work. The participating children had special needs, some more, some less. Coaches alone are not able to fulfill these needs. Again, this has a positive and a negative side concerning the evaluation. The presence of the students was highly valued and the conditions were almost optimal. This could be a way sport (and other activities) should be organized to improve inclusive interactions which lead to positive experiences for all. The negative side is that this is not easily achieved. In a time-limited, university study, it can be manageable, but to the best of our knowledge, there are no studies dealing with a real-life scenario in mainstream sport. There are some studies in Physical Education (PE), which advocate co-teaching or the use of paraeducators, who work together with the PE teachers [38]. Although co-teaching seems to be a good approach [44], a high level of knowledge and the possibility to reflect on the advantages and disadvantages is mandatory [45].

This leads to one of the critical aspects that the clubs mentioned. The training needs of the coaches and other professional staff in the clubs has to be improved. As the coaches are mostly working on a voluntarily basis, it is difficult to force them to get more qualifications. However, maybe that is not really the problem, as most of the coaches are willing to continue their education. But there are few possibilities to do so. Training for coaches is still very concentrated on sport-specific aspects and topics. Inclusion, heterogeneity, and disability are often not mentioned, or only as an additional not mandatory offer, at least in the German Olympic Sports Confederation, which is responsible for most of the training for sport coaches. A cooperation with sport organizations from the disability sector, like Special Olympics, and their experience with inclusive sport is only at the beginning at present. A further deepening of the cooperation would be desirable.

Surprisingly for the club representatives, the attitude of the non-disabled members was no problem at all. At t1, concerns were made about the acceptance of the children with ID by the members and there was some fear of losing old members because of this. This did not happen. We did not ask the members, so we can only speculate about the reasons. Maybe kids at this age are really more open than older people, as has been proposed in several studies [46,47,48]. Maybe the construction of the study itself, especially the engagement of the student facilitators, supported this development. Whatever the reason, it seems possible to include kids at this age in mainstream sport, if enough support is provided and the settings are well prepared.

Future studies could address some limitations of the project. We considered all meanings, thoughts, and wishes from all stakeholders through our very open research approach. The result was a wide array of topics, which were structured through thematic analysis. Nevertheless, some aspects might have been under- or overestimated and repeating the study with other clubs and participants would be valuable. It was not our intention to examine a pre-post-design in a statistical way. The use of questionnaires and scales to measure satisfaction with the program in a more structured way could add some more information. In the present study, the focus was on understanding the processes and the stakeholder’s common experiences of inclusion. Specific outcomes or intra-group comparisons were not the focus of interest. However, this could and should be a topic for future research.

## 5. Conclusions

Answering the question if the program was a success or not is difficult. Relying on the hard facts, may be disappointing. Only four new permanent members were recruited through the project. The parents still did not believe that mainstream clubs can satisfy the needs of their kids, or they did not feel comfortable with the current arrangements in the clubs: for instance, the qualification of coaches, accessibility of facilities, and transport problems. The implementation itself, on the other hand, was judged very good. It was very well appreciated by parents and the children that sport clubs offer caretaking as well as respect among all the participants with a strong reliance on the assistance for the facilitators and setting up of this special program.

The clubs’ points of view are differentiated as well. They are aware of the things that have to be improved and that they are not as far forward as they wish to be. Inclusive sports require a lot of preparation and attention to create good programs, and the needs of all participants must be considered, which is a task for the club management. However, the experience of participating in this study has been an eye-opener for participating clubs. It remains to be hoped that they will transfer these experiences in future actions.

This study, again, adds evidence that some form of co-teaching is essential. One coach on its own will only be successful with very much effort, which cannot be expected from persons who do this job on a voluntary basis. Professional staff, who are paid properly, would be a good possibility, even if this is more a wish than reality.

## Figures and Tables

**Table 1 ijerph-19-11540-t001:** Study process.

Time	Topics
Five weeks prior to start of the program	Information about the program on the website of the agency, time to register
T0	Information letters concerning sport offers, addresses, facilities, and details of the study; consent forms were sent to the families
T1	Start of the sport program, first round of focus groups
T2, eight weeks later	End of the sports program, second round of focus group interviews

**Table 2 ijerph-19-11540-t002:** Interview topics for qualitative program evaluation, divided by interview groups at t1.

Group	Topics
parents and children	1. Importance of sport in everyday life 2. Motivation to join the program 3. Experiences in mainstream sport (positive and negative) 4. Hopes and wishes for the sport program
Clubs	1. Motivation to join the program 2. Experiences with people with ID in their clubs 3. Expected benefit of including people with ID 4. Expected barriers of including people with ID

**Table 3 ijerph-19-11540-t003:** Interview questions for qualitative program evaluation, divided by interview groups at t2.

Group	Questions
parents and children	1. How was the overall experience of taking part in the program? 2. What was good, what has to be improved? 3. Did you feel welcomed? 4. Were there any problems/barriers? 5. Did your kids/you find friends? 6. Will your kids/you stay in your sport group(s)?
clubs	1. How was the overall experience of taking part in the program? 2. What was good, what has to be improved? 3. Were there any problems/barriers? 4. What did your other members say? 5. What was the feedback of the coaches? 6. Will you offer inclusive groups in the future?

## Data Availability

The data reported in this study are available on reasonable request to the corresponding author.

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
