# Peer review of "“Sports for All”—An Evaluation of a Community Based Physical Activity Program on the Access to Mainstream Sport for Children with Intellectual Disability"

_ijerph, 2022, doi:10.3390/ijerph191811540_

Round 1
Reviewer 1 Report
--General comments
The manuscript deals with an important theme, but its methods and results are likely to not be reproduced. This occurs because the instruments did not sustain any statistical possibility. Based on this problem, there is no way to form subgroups (sex, age, or social condition, for instance). Without comparable data, the manuscript only provides “portraits” of the program, and not its success or further inferences. The sentence “Answering the question if the program was a success or not is difficult” inserted in the limitation’s topic reflects my critics. Several sentences lack scientific soundness (e.g We tried our best). The introduction is confusing, imitating the reader to reach the objectives.
--Specific comments
Introduction
- Be consistent with the PA acronym after its insertion. The same for ID.
- Lines 30-32 - What is the difference among all these variables?
- Lines 44-49 – The entire paragraph is confusing. I did not understand the similarity between “organized sport, club and facilities”.
- Line 50 – state of research?
- Line 50-51 – Authors are probably misleading some fundamentals here and thorough the introduction. Can people improve their PA by practicing some sport?
Overall, the introduction can be improved. I recommend limiting it to four or five paragraphs. In each paragraph the idea must be introduced and partially finalized, providing a link to the next one. The three first sentences of the first paragraph are an example. The first sentence provides information on PA. In the second you incorporate the people with ID, and then, in the third, you return with the PA. One possibility is 1º paragraph - introduce PA, and its importance for health and other outcomes; 2º paragraph – people with ID and the relevance of PA for this population; (link to the motives and barriers); 3ª paragraph – the lack of research in the context; 4º paragraph – the main questions to be answered with the study. Be careful with the terms, and avoid long sentences with several prepositions.
Material and Methods
Why families from low-THRESHOLD communities were recruited? What is the impact of the social condition on your outcomes?
Lines 131-145 – Some well-designed organograms can improve the experimental design and drop-outs (with reasons).
Results
Although I understand the non-numeric essence of thematic analysis, I missed some questionnaires with scores. Based on this limitation, you need to provide insight for future studies in the discussion.
Discussion
The objectives must be presented in the introduction, and not in this section. Before methods and results, the readers need to know the study’s aims. Also, it is not necessary to bring the objectives with the terms “the first was…..the second objective was…and so on”.
Reviewer 2 Report
.

Round 2
Reviewer 1 Report
The authors devoted great efforts and responded to all my suggestions or criticisms. I understand that within its proposal, the study presents good quality. My last recommendation is to synthesize the discussion in a single paragraph, leaving the message clear about the main findings of the study. In this sense, it is more elegant to insert the limitations in the last paragraph of the discussion, leaving the conclusion separate and summarized.
Author Response
I am very happy, that you were satisfied with my revisions, they made the manuscript much better! Thank you for your support! I have adressed your last recommendation by changing the order of limitations, discussion and conclusion following your advice.
Reviewer 2 Report
Thank you for your work.
Author Response
I am happy, that you were satisfied with my revision! As far as i can see, you have no recommendations for a further revision.
Thank you for your support, it has made the manuscript much better now!